# Prevalence and determinants of non-communicable diseases risk factors among reproductive-aged women: Findings from a nationwide survey in Bangladesh

**Saifur Rahman Chowdhury**[1,2]*, **Md. Nazrul Islam**[2,3], **Tasbeen Akhtar Sheekha**[3], **Shirmin Bintay Kader**[3], **Ahmed Hossain**[4,5]

1 Department of Health Research Methods, Evidence, and Impact (HEI), McMaster University, Hamilton, Ontario, Canada, 2 Department of Public Health, North South University, Dhaka, Bangladesh, 3 Department of Community Health and Epidemiology, College of Medicine, University of Saskatchewan, Saskatoon, SK, Canada, 4 Health Services Administration, College of Health Sciences, University of Sharjah, Sharjah, United Arab Emirates, 5 Global Health Institute, North South University, Dhaka, Bangladesh

* saifur@mcmaster.ca, saifur.rahm1994@gmail.com

**Data Availability Statement:** All relevant data are within the manuscript and its Supporting Information files.

## Abstract

### Introduction

Knowing the risk factors like smoking status, overweight/obesity, and hypertension among women of reproductive age could allow the development of an effective strategy for reducing the burden of non-noncommunicable diseases. We sought to determine the prevalence and determinants of smoking status, overweight/obesity, hypertension, and cluster of these non-noncommunicable diseases risk factors among Bangladeshi women of reproductive age.

### Methods

This study utilized the Bangladesh Demographic and Health Survey (BDHS) data from 2017–2018 and analyzed 5,624 women of reproductive age (age 18–49 years). This nationally representative cross-sectional survey utilized a stratified, two-stage sample of households. Poisson regression models with robust error variance were fitted to find the adjusted prevalence ratio (APR) for smoking, overweight/obesity, hypertension, and for the clustering of non-noncommunicable diseases risk factors across demographic variables.

### Results

The average age of 5,624 participants was 31 years (SD = 9.1). The prevalence of smoking, overweight/obesity, and hypertension was 9.6%, 31.6%, and 20.3%, respectively. More than one-third of the participants (34.6%) had one non-noncommunicable diseases risk factor, and 12.5% of participants had two non-noncommunicable diseases risk factors. Age, education, wealth index, and geographic location were significantly associated with smoking status, overweight/obesity, and hypertension. Women between 40–49 years had more non-noncommunicable diseases risk factors than 18–29 years aged women (APR: 2.44; 95% CI: 2.22–2.68). Women with no education (APR: 1.15; 95% CI: 1.00–1.33), married (APR:

**Funding:** The author(s) received no specific funding for this work.

**Competing interests:** The authors have declared that no competing interests exist.

2.32; 95% CI: 1.78–3.04), and widowed/divorced (APR: 2.14; 95% CI: 1.59–2.89) were more likely to experience multiple non-noncommunicable diseases risk factors. Individuals in the Barishal division, a coastal region (APR: 1.44; 95% CI: 1.28–1.63) were living with a higher number of risk factors for non-noncommunicable diseases than those in the Dhaka division, the capital of the country. Women who belonged to the richest wealth quintile (APR: 1.82; 95% CI: 1.60–2.07) were more likely to have the risk factors of non-noncommunicable diseases.

## Conclusions

The study showed that non-noncommunicable diseases risk factors are more prevalent among women from older age group, currently married and widowed/divorced group, and the wealthiest socio-economic group. Women with higher levels of education were more likely to engage in healthy behaviors and found to have less non-noncommunicable diseases risk factors. Overall, the prevalence and determinants of non-noncommunicable diseases risk factors among reproductive women in Bangladesh highlight the need for targeted public health interventions to increase opportunities for physical activity and reduce the use of tobacco, especially the need for immediate interventions in the coastal region.

## Introduction

The major cause of global death and disability is non-communicable diseases (NCDs) [1, 2]. The most common NCDs are cardiovascular diseases (CVD), cancer, chronic respiratory diseases, and diabetes [2]. In 2022, World Health Organization (WHO) reported that each year 17 million people die from an NCD before age 70; an estimated 86% of these premature deaths occur in low- and middle-income countries (LMICs), and over 80% of these deaths occur due to the above-mentioned NCDs [2].

In Bangladesh, NCDs are significant cause of mortality, responsible for 572,600 deaths, 67% of total deaths in 2016 [3]. The prevalence of NCDs has increased over the last twenty years, and it is predicted to increase more as Bangladesh is passing through a growing stage of epidemiological transition [1, 4–6]. In the year 2000, NCDs were responsible for 43% of all deaths in Bangladesh [7]. This number grew to 59% in 2010 and then reached an alarming 70% in 2019 [7]. Bangladesh surfaced as a lower-middle-income country with an experience of rapid growth in economy and urbanization in the previous decades [8]. As a result of these socio-economic changes, a large number of people are leading a more sedentary lifestyle influenced by the change in dietary habits, increased supply and demand for unhealthy processed food, and physical inactivity followed by irregularities in mealtimes, smoking, and alcohol consumption [6, 8–10]. Moreover, there are metabolic risk factors that contribute to the development of NCDs [11]. The most common metabolic risk factors are obesity, high blood sugar, high blood pressure, and increased cholesterol [5]. There is an increased chance of developing NCDs when two or more of these factors combine, referred to as clustering of risk factors [12]. Consequently, a considerable proportion of the population aged 18 years and above suffered from being overweight (29.4%) [13], hypertension (26.2%) [14], and diabetes (9.2%) [15] in Bangladesh. In two meta-analyses that included studies up to April 2017, the weighted pooled prevalence of CVD was estimated at 5% [6], and the prevalence of metabolic syndrome and high fasting glucose were 30.0% and 28%, respectively [5].

Women are at an increased risk of developing NCDs as they mostly experience a combination of multiple behavioral and metabolic risk factors [16, 17]. According to WHO, 94% of all maternal deaths occur in LMICs, and significant causes of these deaths are pre-existing medical conditions and risk factors [18]. Evidence showed that between 1992 and 2015, the prevalence of diabetes had a threefold to fourfold increase, with a prediction to reach around 24% in men and 33.5% in women by 2030, with higher odds of multimorbidity than men [19, 20]. Women's fetal health and reproductive health become severely affected by the NCDs risk factors [21, 22]. For example, hypertension in pregnancy increases menstrual problems, complications in pre- and post-natal periods, and maternal mortality [23]. Moreover, previous research showed that women were more likely to be overweight in urban areas because of rapid urbanization, whereas they were more likely to be underweight in rural areas because of inadequate provision of healthy food [24]. Lack of proper nutrition may lead to serious metabolic changes contributing to the development of heart disease, hypertension, and diabetes among them as well as among their future generation [9]. In addition, obesity is related to polycystic ovary syndrome [25], and hypertension is the common cause of higher maternal mortality and stillbirth [23]. However, women are less likely to be identified or treated as they demonstrate fewer signs and symptoms of NCDs, for instance, of CVD, compared to men and get a lack of attention in case of disease prevention strategies [26].

Although Bangladesh has been successful in achieving several Sustainable Development Goals (SDG), the government and many international organizations are still working to decrease the burden of NCDs, as these are considered a challenge in the field of public health. It is evident that to prevent and control NCDs, the socio-demographic determinants of major NCDs risk factors and their clustering should be explored and understood in a systematic way. There were many studies that focused on the prevalence and risk factors of NCDs [1, 4–6, 10, 12], in addition to diabetes, hypertension [1, 14, 15, 27], overweight and obesity [13]. Though a large number of studies were nationally representative studies and few studies used earlier data or focused on a specific geographic region, studies on women of reproductive age addressing the overall social determinants and prevalence of risk factors of NCDs were very rare. This study, therefore, aimed to investigate the prevalence of risk factors of NCDs and determine their socio-demographic distributions among Bangladeshi reproductive-aged women.

## Materials and methods

### Study population and data source

We used the data from the latest Bangladesh Demographic and Health Survey (BDHS) 2017–2018 to explore the prevalence of NCDs risk factors among reproductive-aged women. The BDHS is a nationally representative cross-sectional survey that covers the entire population residing in non-institutional dwelling units in the country. The BDHS survey was performed using a stratified, two-stage household sample. In the first step, 675 enumeration areas (EAs) (250 in urban areas and 425 in rural areas) were picked with a probability proportional to the size of the EA. In the second round of sampling, an average of 30 households per EA were gathered independently from urban and rural areas and from each of the eight divisions in order to obtain statistically reliable estimates of key demographic and health factors for the entire country. The 2017–18 BDHS used six types of questionnaires: (1) the Household Questionnaire, (2) the Woman's Questionnaire (completed by ever-married women aged 15–49), (3) the Biomarker Questionnaire, (4) two Verbal Autopsy Questionnaires to collect data on causes of death among children under age 5, (5) the Community Questionnaire, and (6) the Fieldworker Questionnaire. The first three questionnaires were based on the model

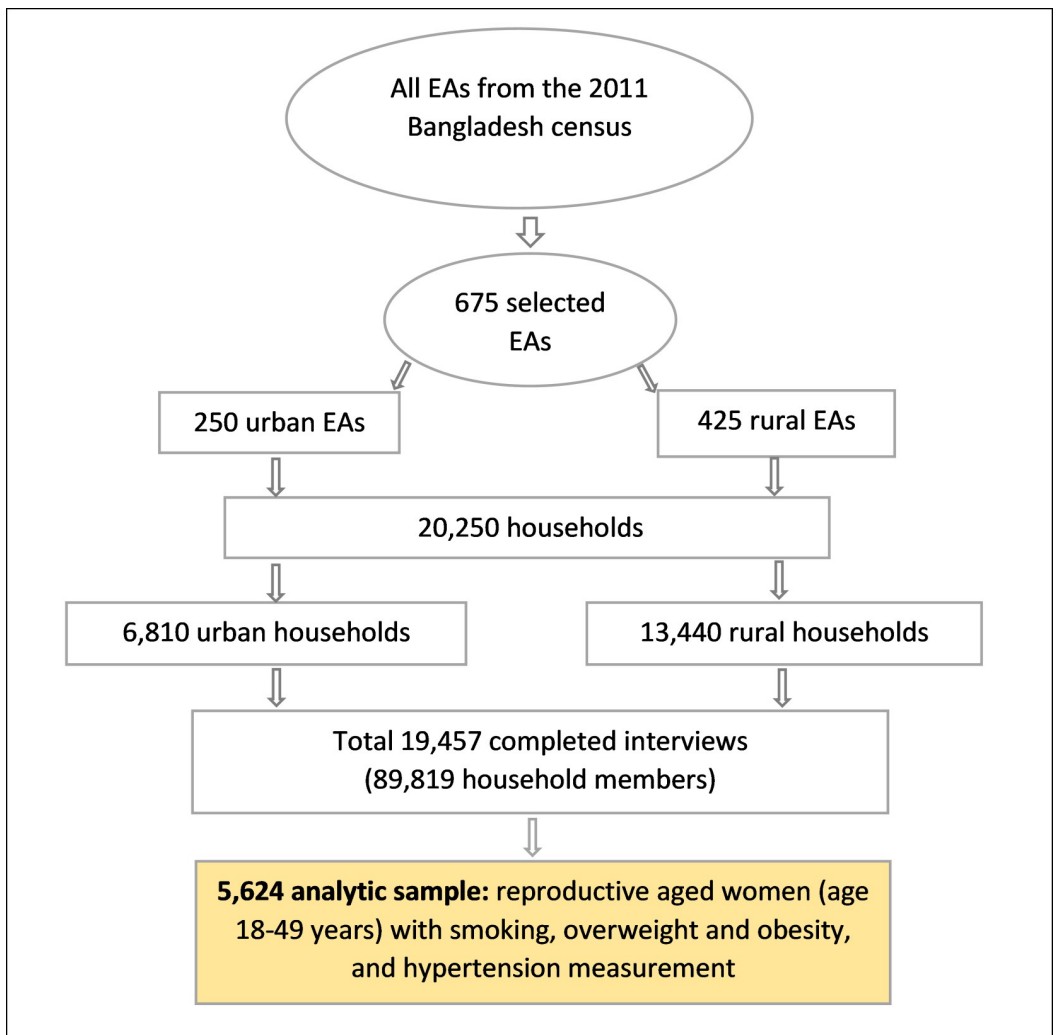

**Fig 1. Schematic representation of the sampling procedure of the Bangladesh Demographic and Health Survey, 2017–2018.**

questionnaires developed for the DHS-7 Program, adapted to the situation and needs in Bangladesh and taking into account the content of the instruments employed in prior BDHS surveys. The questionnaires were developed in English and then translated into and printed in Bangla. Back translations were conducted by people not involved with the Bangla translations. Detailed survey sampling and the data collection procedure have been published in the BDHS survey report [28]. On the survey, 20,250 households were selected, interviews were completed in 19,457 households with an overall 96.5% household response rate, and information was recorded from 89,819 household members [28]. Finally, we included 5,624 reproductive-aged women (age 18–49 years) who had their blood pressure, height and weight, and smoking status recorded (**Fig 1**).

## Outcome variables

The outcome variables for this research were the three risk factors for NCDs: smoking, overweight and obesity, and hypertension. These three selected NCDs risk factors are available in

the BDHS 2017–2018 survey data set. Moreover, we defined the clustering of these three risk factors and assessed the prevalence and associated factors.

## Smoking

In the BDHS survey, participants' current smoking status was recorded as a dichotomous response.

## Overweight and obesity

Height and weight measures were collected for respondents in all of the households sampled. Weight measurements were obtained using an electronic scale (model number SECA 878U) with a digital screen, and height measurements were carried out with a measuring board (ShorrBoard). The health technician and one female technician were deployed to take both measurements. Women who were pregnant on the day of the survey visit or had given birth during the preceding two months were excluded. Body mass index (BMI) was calculated by dividing weight in kilograms by height in meters squared ($kg/m^2$). The overweight was considered for the individual with BMI between 25.0 and 29.9 $kg/m^2$, and the obese was BMI greater than or equal to 30.0 $kg/m^2$ [28]. We combined overweight and obese to make a single binary variable and defined an individual in the overweight and obese category if the BMI was greater than or equal to 25.0 $kg/m^2$.

## Hypertension

Blood pressure was measured for all participants aged 18 and above in the subsample of one-fourth of the households. All participants who qualified for blood pressure measures were contacted and instructed on the procedure. Those who gave their consent were measured for their blood pressure. The automatic device includes separate cuffs BP monitor to measure blood pressure. Three blood pressure measurements were taken at intervals of approximately 10 minutes. Typically, the average of the second and third measurements was utilized to reflect the blood pressure results of respondents. Individuals were diagnosed having hypertension if they had an average systolic blood pressure level of 140 mmHg or above, or diastolic blood pressure level of 90 mmHg or above, or they were taking antihypertensive medication at the time of the survey [29, 30].

## Clustering of NCDs risk factors

The clustering of NCDs risk factors was defined as the existence of multiple risk factors in each participant. The number of risk factors from smoking, overweight/obesity, or hypertension was counted to assess the clustering of risk factors. Therefore, their existence is deemed to range from 0 to 3.

## Explanatory variables

We considered the socio-demographic covariates related to smoking, overweight/obesity, and hypertension. These included the respondents' ages (18–29, 30–39, and 40–49 years), education (no education, primary, secondary, and higher-level), marital status (married, never married, and widowed/divorced), residence (rural, urban), geographical areas (Barishal, Chattogram, Dhaka, Khulna, Mymensingh, Rajshahi, Rangpur, and Sylhet divisions), occupational status (not working, service, agriculture/self-employed, and manual), wealth index (poorest, poorer, middle, richer and richest). According to the BDHS report, the wealth index of the households was calculated by the principal component analysis. The scores were given based on available

consumer goods in the household (television, bicycle, car, etc.), and household characteristics (source of drinking water, sanitation facilities, flooring materials, etc.). Then the national wealth quintiles were compiled by assigning the household score to each usual household member, ranking each person in the household population by her/his score, and then dividing the distribution into five equal categories, each comprising 20% of the population [28].

## Data analysis

The socio-demographic characteristics of the respondents were reported in mean and weighted percentages. The crude prevalence of smoking, overweight and obesity, and hypertension were determined after accounting for the complex survey design and survey sampling weights. Differences between categorical variables were tested using the chi-square tests. We included variables in the multivariable models if the chi-square test p-value <0.20 or based on their clinical relevance. We applied modified Poisson regression models with robust error variance to identify factors associated with NCDs risk factors (smoking, overweight/obesity, and hypertension) and reported the results as adjusted prevalence ratio (APR) with a 95% confidence interval (CI). We used this model since the odds ratio estimated using logistic regression from a cross-sectional study may significantly overestimate relative risk when the outcome is common [31, 32]. Secondly, in the case of convergence failure with the log-binomial model, modified Poisson regression with a robust error variance performs better in estimating the prevalence ratio from a cross-sectional study [33, 34]. The modified Poisson regression model with a robust error variance was utilized previously as an alternative to the logistic regression model if the prevalence of binary outcome is more than >10% [35–37]. Finally, the number of risk factors present within each participant (from 0 to 3) was counted to assess the clustering of risk factors and analyzed using the Poisson regression model with robust error variance. We used the robust error variance option to obtain robust standard errors for the parameter estimates as recommended by Cameron and Trivedi (2022) to control for mild violations of underlying assumptions [38]. All statistical tests were two-tailed, and a p-value of <0.05 was regarded as statistically significant. Statistical software STATA-16 (Stata Corp LP, College Station, TX, USA) was used to conduct the analysis. Since the BDHS survey was based on the two-stage stratified cluster sampling technique, recommended sample weights provided by the BDHS 2017–18 were used for the analyses [28]. So, to account for the complex survey design, we considered the sample weights, primary sampling units, and strata using "SVY" command of STATA.

## Ethical considerations

We used secondary data from the country representative survey (BDHS 2017–2018). The study protocol for BDHS was approved by the ICF (international institutional review board), and the data is publicly available (http://dhsprogram.com/data/available-datasets.cfm). Therefore, no further ethical approval was necessary for this study. However, we received authorization from the DHS to use the datasets. Informed consent was obtained from each participant of the survey.

Before enrolling in the survey by using the Introduction and Consent form of the survey. It was also explained that the information would be kept strictly confidential and not be shared with anyone except the survey team members.

## Results

### Socio-demographic characteristics of study participants

The socio-demographic characteristics of the study participants (n = 5,624) are shown in **Table 1**. The mean age of research participants was 31 years (standard deviation, SD = 9.1).

**Table 1. Socio-demographic characteristics of participants.**

| Variables | Un-weighted count | Weighted percent/Mean (SD) |
|---|---|---|
| **Total** | 5,624 | 100.00 |
| **Mean age (year)** | | 31.0 (9.1) |
| **Age group (year)** | | |
| 18–29 | 2,689 | 48.3 |
| 30–39 | 1,716 | 30.3 |
| 40–49 | 1,219 | 21.5 |
| **Educational status** | | |
| No education | 957 | 17.8 |
| Primary | 1,691 | 29.9 |
| Secondary | 2,051 | 37.7 |
| Higher | 925 | 14.7 |
| **Marital status** | | |
| Never married | 402 | 6.3 |
| Married | 4,902 | 88.2 |
| Widowed/divorced | 320 | 5.5 |
| **Residence** | | |
| Rural | 3,533 | 71.8 |
| Urban | 2,091 | 28.2 |
| **Division** | | |
| Barishal | 590 | 5.5 |
| Chattogram | 827 | 18.6 |
| Dhaka | 799 | 24.4 |
| Khulna | 729 | 11.7 |
| Mymensingh | 589 | 7.5 |
| Rajshahi | 732 | 14.2 |
| Rangpur | 690 | 11.7 |
| Sylhet | 668 | 6.5 |
| **Wealth index** | | |
| Poorest | 1,069 | 19.0 |
| Poorer | 1,051 | 19.5 |
| Middle | 1,081 | 19.9 |
| Richer | 1,091 | 19.6 |
| Richest | 1,332 | 22.0 |
| **Occupational status*** | | |
| Not working | 3,066 | 54.4 |
| Services | 242 | 4.0 |
| Agriculture/self-employed | 1,667 | 30.2 |
| Manual | 641 | 11.3 |

*8 cases missing

Approximately half (48.3%) of them were between 18 and 29 years old. In terms of educational attainment, 17.8% of the participants did not have any formal education. On the other hand, only 14.7% of respondents had education above the secondary level. In this study, 88.2% of respondents were married. More than two-thirds (71.8%) of the participants resided in rural areas. However, about one-quarter (24.4%) of the participants came from the Dhaka division, which is the capital of Bangladesh. The richest comprised 22% of the total participants based

on the wealth index. Surprisingly, the proportions of participants between the poorest and richer quintiles were almost identical. More than half (54.4%) of the participants were not involved in any formal work.

## Prevalence of non-communicable diseases risk factors among reproductive-aged women

Table 2 depicts the prevalence of risk factors for NCDs among reproductive-aged women. The prevalence of smoking was 9.6% among the study participants. Smoking was more prevalent among the age group 40–49 years (19.3%), women with no educational status (19.7%), Sylhet division residents (21.7%), participants with the poorest wealth index (14%), and the agriculture/self-employed women (14.5%) (p <0.001). Compared to other risk factors, overweight and obesity were profound among the participants (31.6%). About 40.3% of the participants between 30–39 years were either overweight or obese (p <0.001). Likewise, overweight and obesity were more prevalent among married (33.4%), urban residents (40.2%), Chattogram division residents (36.3%), the richest (52.3%), and service holders (35.2%) (p <0.001). In this study, 20.3% of women were suffering from hypertension. Women aged 40 to 49 had the highest prevalence of hypertension (39.2%) (p <0.001). In addition, the prevalence of hypertension was higher among women who were uneducated (29.1%), and divorced/widowed (25.7%) (p <0.001).

Furthermore, we determined the prevalence of NCDs risk factors by the number of factors across different age groups (Fig 2). More than one-third of the participants had one NCDs risk factor, and 12.5% of participants had two NCDs risk factors.

## Association between socio-demographic characteristics and non-communicable diseases risks factors

According to Table 3, the modified Poisson regression models revealed the adjusted association between socio-demographic characteristics and non-communicable diseases (NCDs) risk factors. The participants' age was strongly associated with the risk factors for NCDs. Compared to participants aged 18–29 years, those aged 30–39 years (APR: 1.83; 95% CI: 1.45–2.30) and 40–49 years (APR: 2.65; 95% CI: 2.07–3.38) were more likely to be smokers. The prevalence ratio for overweight and obesity in the age group 30–39 years (APR: 1.71; 95% CI: 1.55–1.89) and 40–49 years (APR: 1.77; 95% CI: 1.57–1.98) were significantly higher than the age group18-29 years. Similarly, the prevalence of hypertension was significantly higher in the 30–39 years (APR: 2.58; 95% CI: 2.18–3.05) and 40–49 years (APR: 3.94; 95% CI: 3.34–4.69) age groups compared to the younger age group. The prevalence ratio of smoking was almost 3 times higher among participants with no education (APR: 3.19; 95% CI: 1.85–5.49) compared to those with a higher level of education. Married women (APR: 2.71; 95% CI: 1.96–3.75) and widowed/divorced women (APR: 2.16; 95% CI: 1.49–3.15) demonstrated a higher prevalence of overweight and obesity compared to the never-married women.

For current smoking, the prevalence ratio of the participants from Barishal (APR: 2.81; 95% CI: 2.04–3.86), Mymensingh (APR: 2.06; 95% CI: 1.47–2.90), and Sylhet (APR: 3.07; 95% CI: 2.26–4.16) was comparatively higher than the participants from Dhaka, the capital of Bangladesh. However, hypertension was almost 1.5 times higher among the participants from Barishal, Chattogram, and Rangpur than the participants from Dhaka. In addition, individuals with hypertension (APR: 1.51; 95% CI: 1.22–1.87), overweight and obesity (APR: 2.88; 95% CI: 2.39–3.47) were more likely to belong to the wealthiest quintile. Compared to those who were not employed, those whose occupations involved agriculture or self-employment had a higher prevalence ratio for current smoking (APR: 1.42; 95% CI: 1.18–1.71).

**Table 2. Prevalence of non-communicable diseases risk factors among reproductive-aged women.**

| Variables | Current smoking | | Overweight and obesity | | Hypertension | |
|---|---|---|---|---|---|---|
| | n | Prevalence, % (95% CI) | n | Prevalence, % (95% CI) | n | Prevalence, % (95% CI) |
| **Total** | 5,624 | 9.6 (8.7–10.5) | 5,603 | 31.6 (30.2–33.0) | 5,624 | 20.3 (19.1–21.5) |
| **Age group (year)** | | | | | | |
| 18–29 | 2,689 | 4.3 (3.5–5.2) | 2,679 | 22.9 (21.1–24.8) | 2,689 | 9.0 (7.9–10.3) |
| 30–39 | 1,716 | 11.1 (9.5–12.9) | 1,708 | 40.3 (37.6–42.9) | 1,716 | 24.8 (22.5–27.2) |
| 40–49 | 1,219 | 19.3 (16.7–22.1) | 1,216 | 38.8 (35.9–41.8) | 1,219 | 39.2 (36.1–42.3) |
| *P-value* | | <0.001 | | <0.001 | | <0.001 |
| **Educational status** | | | | | | |
| No education | 957 | 19.7 (17.1–22.6) | 952 | 26.1 (23.3–29.2) | 957 | 29.1 (25.9–32.4) |
| Primary | 1,691 | 12.6 (11.0–14.4) | 1,688 | 30.8 (28.4–33.3) | 1,691 | 22.3 (20.2–24.6) |
| Secondary | 2,051 | 5.1 (4.1–6.1) | 2,044 | 33.8 (31.4–36.3) | 2,051 | 17.1 (15.4–19.0) |
| Higher | 925 | 2.6 (1.7–4.1) | 919 | 34.1 (30.4–37.9) | 925 | 13.6 (11.3–16.2) |
| *P-value* | | <0.001 | | 0.001 | | <0.001 |
| **Marital status** | | | | | | |
| Never married | 402 | 1.4 (0.5–3.6) | 396 | 10.4 (7.5–14.4) | 402 | 6.18 (4.01–9.40) |
| Married | 4,902 | 9.7 (8.8–10.7) | 4,890 | 33.4 (31.9–34.9) | 4,902 | 20.95 (19.69–22.27) |
| Widowed/divorced | 320 | 16.4 (2.4–21.4) | 317 | 26.9 (21.6–32.9) | 320 | 25.7 (20.6–31.6) |
| *P-value* | | <0.001 | | <0.001 | | <0.001 |
| **Residence** | | | | | | |
| Rural | 3,533 | 10.2 (9.1–11.4) | 3,519 | 28.2 (26.5–29.9) | 3,533 | 19.9 (18.5–21.4) |
| Urban | 2,091 | 7.9 (6.5–9.6) | 2,084 | 40.2 (37.6–42.8) | 2,091 | 21.2 (19.2–23.3) |
| *P-value* | | 0.024 | | <0.001 | | 0.317 |
| **Division** | | | | | | |
| Barishal | 590 | 21.3 (17.7–25.5) | 588 | 29.3 (25.1–33.8) | 590 | 24.2 (20.4–28.5) |
| Chattogram | 827 | 8.6 (6.6–11.1) | 824 | 36.3 (32.6–40.1) | 827 | 23.5 (20.7–26.6) |
| Dhaka | 799 | 6.4 (4.8–8.4) | 795 | 35.7 (32.4–39.2) | 799 | 16.9 (14.4–19.8) |
| Khulna | 729 | 6.8 (4.8–9.7) | 728 | 35.5 (31.8–39.3) | 729 | 20.5 (17.3–24.1) |
| Mymensingh | 589 | 17.0 (13.7–20.9) | 587 | 24.7 (21.0–28.8) | 589 | 15.8 (12.9–19.1) |
| Rajshahi | 732 | 5.1 (3.2–7.8) | 728 | 28.1 (24.6–31.9) | 732 | 20.6 (17.4–24.3) |
| Rangpur | 690 | 8.9 (6.1–12.9) | 689 | 25.2 (21.9–28.8) | 690 | 23.4 (20.5–26.5) |
| Sylhet | 668 | 21.7 (17.7–26.2) | 664 | 24.4 (20.1–29.3) | 668 | 18.9 (15.8–22.6) |
| *P-value* | | <0.001 | | <0.001 | | 0.001 |
| **Wealth index** | | | | | | |
| Poorest | 1,069 | 14.0 (11.7–16.8) | 1,066 | 16.4 (14.1–19.2) | 1,069 | 17.3 (14.8–20.3) |
| Poorer | 1,051 | 12.5 (10.4–14.8) | 1,046 | 21.7 (19.1–24.6) | 1,051 | 18.8 (16.4–21.5) |
| Middle | 1,081 | 8.4 (6.8–10.3) | 1,080 | 29.8 (26.8–33.0) | 1,081 | 20.1 (17.7–22.8) |
| Richer | 1,091 | 8.4 (6.8–10.4) | 1,088 | 34.6 (31.5–37.8) | 1,091 | 21.4 (18.8–24.2) |
| Richest | 1,332 | 5.1 (4.0–6.7) | 1,323 | 52.3 (49.2–55.4) | 1,332 | 23.2 (20.8–25.9) |
| *P-value* | | <0.001 | | <0.001 | | 0.019 |
| **Occupational status*** | | | | | | |
| Not working | 3,066 | 7.5 (6.5–8.7) | 3,052 | 34.4 (32.4–36.3) | 3,066 | 20.2 (18.7–21.8) |
| Services | 242 | 5.6 (3.2–9.9) | 238 | 35.2 (28.5–42.5) | 242 | 20.8 (15.5–27.2) |
| Agriculture/self-employed | 1,667 | 14.5 (12.6–16.6) | 1,667 | 26.0 (23.9–28.2) | 1,667 | 22.3 (20.0–24.6) |
| Manual | 641 | 7.6 (5.3–10.7) | 638 | 32.5 (28.5–36.7) | 641 | 15.6 (12.8–18.8) |
| *P-value* | | <0.001 | | <0.001 | | 0.013 |

*8 cases missing

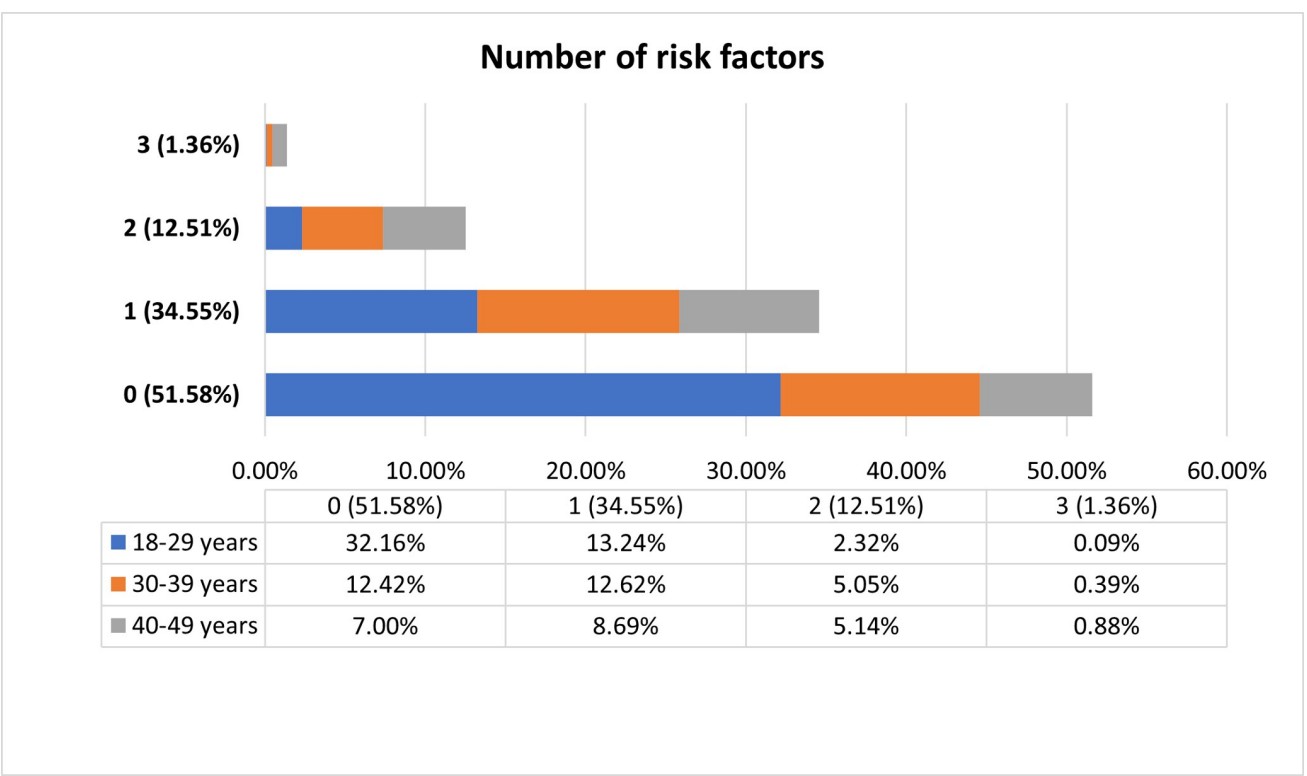

**Fig 2. Prevalence of number of non-communicable diseases risk factors among reproductive-aged women.**

## Mean number of NCDs risk factors and multivariable analysis of clustering of NCDs risk factors

We examined the clustering of NCDs risk factors (the number of risk factors present in an individual from 0 to 3) in our sample. We performed Poisson regression model with robust error variance. Table 4 shows that women of 40–49 years were more likely to experience multiple non-communicable diseases (NCDs) risk factors than 18–29 years aged women (ARR: 2.44; 95% CI: 2.22–2.68). Women with no education were at slightly higher risk for multiple NCDs risk factors (ARR: 1.15; 95% CI: 1.00–1.33) compared to the women with a higher level of education. In comparison with the never-married women, married (ARR: 2.32; 95% CI: 1.78–3.04), and widowed/divorced (ARR: 2.14; 95% CI: 1.59–2.89) were more likely to experience multiple NCDs risk factors.

Individuals in the Barishal (ARR: 1.44; 95% CI: 1.28–1.63), Chattogram (ARR: 1.25; 95% CI: 1.12–1.40), and Sylhet divisions (ARR: 1.27; 95% CI: 1.12–1.43) were living with a somewhat higher number of NCDs risk factors compared to the individuals in the Dhaka division. Likewise, women belonged to the middle (ARR: 1.25; 95% CI: 1.11–1.41), richer (ARR: 1.44; 95% CI: 1.27–1.63), and richest wealth quintile (ARR: 1.82; 95% CI: 1.60–2.07) were more likely to have multiple risk factors of NCDs compared to the women in the poorest wealth quintile. The regression prevalence ratio plot also demonstrates the determinants of clustering of NCDs risk factors (Fig 3).

## Discussion

Our study attempted to investigate the prevalence and determinants of NCDs risk factors among reproductive-aged women of Bangladesh by using the most recent nationally representative survey. It is well-established that NCDs are increasing at an alarming rate [2]. Globally,

**Table 3. Association between socio-demographic characteristics and non-communicable diseases risk factors identified from the multivariable modified Poisson regression model with robust error variance.**

| Variables | Current smoking, APR (95% CI) | Overweight and obesity, APR (95% CI) | Hypertension, APR (95% CI) |
|---|---|---|---|
| **Age group (year)** | | | |
| 18–29 | Reference | Reference | Reference |
| 30–39 | 1.83 (1.45–2.30)*** | 1.71 (1.55–1.89) *** | 2.58 (2.18–3.05)*** |
| 40–49 | 2.65 (2.07–3.38)*** | 1.77 (1.57–1.98) *** | 3.94 (3.32–4.69)*** |
| **Educational status** | | | |
| No education | 3.19 (1.85–5.49)*** | 0.81 (0.69–0.97)* | 1.33 (1.04–1.71)* |
| Primary | 2.41 (1.43–4.04) ** | 0.98 (0.85–1.12) | 1.27 (1.01–1.61)* |
| Secondary | 1.45 (0.87–2.43) | 1.04 (0.92–1.17) | 1.23 (0.98–1.53) |
| Higher | Reference | Reference | Reference |
| **Marital status** | | | |
| Never married | Reference | Reference | Reference |
| Married | 2.87 (1.10–7.52)* | 2.71 (1.96–3.75)*** | 1.60 (1.02–2.51)* |
| Widowed/divorced | 3.32 (1.22–9.03)* | 2.16 (1.49–3.15)*** | 1.53 (0.94–2.50) |
| **Residence** | | | |
| Rural | Reference | Reference | Reference |
| Urban | 1.23 (1.00–1.50)* | 1.01 (0.92–1.10) | 1.06 (0.94–1.21) |
| **Division** | | | |
| Barishal | 2.81 (2.04–3.86)*** | 1.07 (0.91–1.26) | 1.50 (1.20–1.87)*** |
| Chattogram | 1.41 (1.00–1.98)* | 1.11 (0.98–1.26) | 1.45 (1.20–1.77)*** |
| Dhaka | Reference | Reference | Reference |
| Khulna | 0.91 (0.62–1.34) | 1.11 (0.97–1.28) | 1.18 (0.96–1.46) |
| Mymensingh | 2.06 (1.47–2.90)*** | 0.93 (0.78–1.11) | 1.02 (0.79–1.31) |
| Rajshahi | 0.61 (0.40–0.92)* | 1.01 (0.87–1.18) | 1.27 (1.02–1.57)* |
| Rangpur | 1.00 (0.68–1.46) | 1.03 (0.87–1.22) | 1.52 (1.23–1.89)*** |
| Sylhet | 3.07 (2.26–4.16)*** | 0.87 (0.74–1.02) | 1.19 (0.96–1.49) |
| **Wealth index** | | | |
| Poorest | Reference | Reference | Reference |
| Poorer | 1.00 (0.81–1.24) | 1.27 (1.05–1.54)* | 1.08 (0.89–1.31) |
| Middle | 0.79 (0.62–1.02) | 1.67 (1.39–2.01)*** | 1.19 (0.98–1.44) |
| Richer | 0.90 (0.68–1.17) | 1.97 (1.64–2.36)*** | 1.36 (1.12–1.67)** |
| Richest | 0.66 (0.46–0.94)* | 2.88 (2.39–3.47)*** | 1.51 (1.22–1.87)*** |
| **Occupational status** | | | |
| Not working | Reference | Reference | Reference |
| Services | 1.12 (0.63–2.00) | 0.94 (0.77–1.14) | 1.04 (0.78–1.38) |
| Agriculture/self-employed | 1.42 (1.18–1.71)*** | 0.88 (0.79–0.98)* | 0.91 (0.80–1.04) |
| Manual | 0.94 (0.69–1.27) | 1.00 (0.87–1.14) | 0.76 (0.62–0.93)** |

APR: Adjusted prevalence ratio

***Significant at p-value < 0.001.

**Significant at p-value < 0.01.

*Significant at p-value < 0.05.

the prevalence of NCDs was reported to be much higher in women in comparison to men [2]. It has been reported that in recent years, the prevalence of NCDs doubled in reproductive-aged women of LMICs [39]. The scenario is not different for Bangladesh as well [40].

We found the smoking, overweight/obesity, and hypertension prevalence were 9.6%, 31.6%, and 20.3%, respectively, among reproductive-aged women. The prevalence of these

**Table 4. Mean number of NCDs risk factors and multivariable analysis (Poisson regression model with robust error variance) of clustering of NCDs risk factors.**

| Variables | Mean number (95% CI) | Clustering of NCDs risk factors, APR (95% CI) |
|---|:---:|:---:|
| **Age group (year)** | | |
| 18–29 | 0.36 (0.34–0.39) | Reference |
| 30–39 | 0.76 (0.72–0.80) | 1.94 (1.78–2.11)*** |
| 40–49 | 0.97 (0.92–1.03) | 2.44 (2.22–2.68)*** |
| **Educational status** | | |
| No education | 0.75 (0.69–0.80) | 1.15 (1.00–1.33)* |
| Primary | 0.66 (0.61–0.70) | 1.13 (1.00–1.29)* |
| Secondary | 0.56 (0.53–0.60) | 1.09 (0.97–1.23) |
| Higher | 0.50 (0.45–0.55) | Reference |
| **Marital status** | | |
| Never married | 0.18 (0.13–0.23) | Reference |
| Married | 0.64 (0.62–0.66) | 2.32 (1.78–3.04)*** |
| Widowed/divorced | 0.69 (0.59–0.78) | 2.14 (1.59–2.89)*** |
| **Residence** | | |
| Rural | 0.58 (0.56–0.61) | Reference |
| Urban | 0.69 (0.65–0.73) | 1.06 (0.99–1.14) |
| **Division** | | |
| Barishal | 0.75 (0.68–0.82) | 1.44 (1.28–1.63)*** |
| Chattogram | 0.68 (0.63–0.74) | 1.25 (1.12–1.40)*** |
| Dhaka | 0.59 (0.53–0.64) | Reference |
| Khulna | 0.63 (0.57–0.69) | 1.10 (0.97–1.24) |
| Mymensingh | 0.58 (0.51–0.64) | 1.13 (0.98–1.30) |
| Rajshahi | 0.54 (0.48–0.60) | 1.02 (0.90–1.16) |
| Rangpur | 0.58 (0.52–0.63) | 1.15 (1.01–1.32)* |
| Sylhet | 0.65 (0.58–0.72) | 1.27 (1.12–1.43)*** |
| **Wealth index** | | |
| Poorest | 0.48 (0.43–0.53) | Reference |
| Poorer | 0.53 (0.48–0.58) | 1.11 (0.99–1.26) |
| Middle | 0.58 (0.54–0.63) | 1.25 (1.11–1.41)*** |
| Richer | 0.64 (0.59–0.69) | 1.44 (1.27–1.63)*** |
| Richest | 0.81 (0.76–0.85) | 1.82 (1.60–2.07)*** |
| **Occupational status** | | |
| Not working | 0.62 (0.59–0.65) | Reference |
| Services | 0.62 (0.50–0.73) | 0.99 (0.83–1.18) |
| Agriculture/self-employed | 0.63 (0.59–0.67) | 0.98 (0.90–1.06) |
| Manual | 0.55 (0.49–0.61) | 0.90 (0.80–1.01) |

APR: Adjusted prevalence ratio

***Significant at p-value < 0.001.

**Significant at p-value < 0.01.

*Significant at p-value < 0.05.

NCDs risk factors is relatively high in the context of the study population. However, the distribution of the NCDs risk factors varies according to the different socio-demographic variables. Our findings were similar to other previous study findings of Bangladesh, which also reported similar prevalence of NCDs risk factors [1, 40–42]. The result of our study showed that the prevalence of smoking increased with age. The findings of our study were in line with the

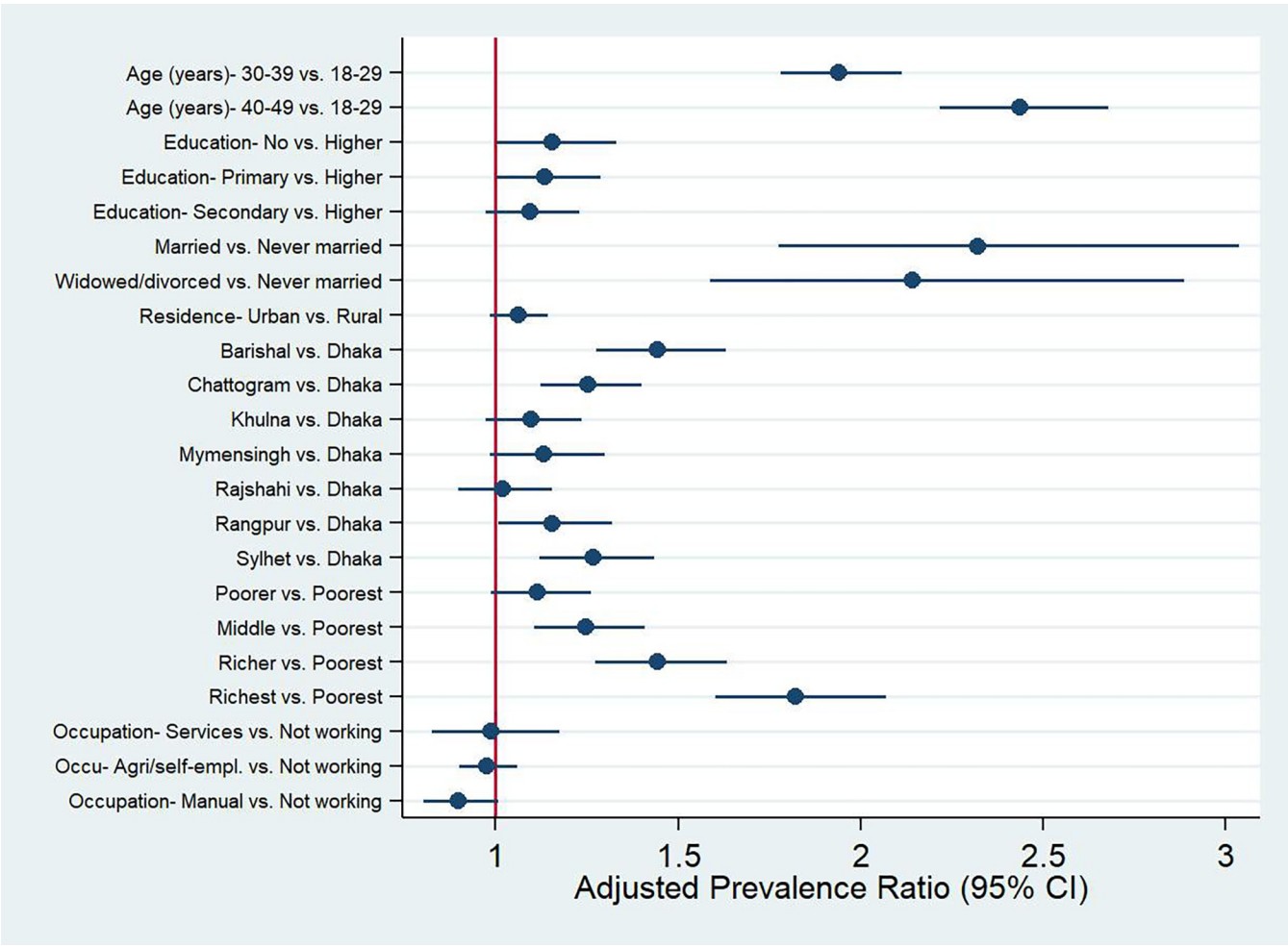

**Fig 3. Regression prevalence ratio plot for the socio-demographic characteristics associated with clustering of NCDs risk factors identified from the Poisson regression model with robust error variance.**

findings of a similar nationwide survey held in Nepal [43]. The high prevalence of smokers was reported commonly among women aged 40+ years, living in a rural area, illiterate, widowed/divorced, and with lower socio-economic status, which is similar to other studies [43–46]. A three-fold higher prevalence of smoking among aged (30–40 years), divorced, and illiterate might be a reflection of coping strategies, and smoking might be an option to overcome their loneliness [47]. Bista et al. found in their study in Nepal that the proportion of tobacco usage was more than 2 times higher among 30–40 years aged women and nearly four times higher among 40+years women in comparison to 15–29 years women [43]. The trend was similar for our study as well. This might be due to the cultural and socio-demographic similarities between Nepal and Bangladesh. Sreeramareddy et al. found similar results while comparing nationwide surveys in nine South-Asian countries, including Bangladesh and Nepal [48]. Respondents from urban areas reported being more smokers. This result is similar to the previous study conducted in Bangladesh [49]. Also, we found participants from Sylhet were more than 3 times higher smoking prevalent than the participants from Dhaka (capital). Similar area-level variation was reported in a previous study conducted in Bangladesh [50]. This difference according to demographics may be due to the fact that in countries with limited

resources, such as Bangladesh, people have a relatively lower level of education and awareness on the diagnosis, care for NCDs, and their consequences.

We observed an increasing trend in overweight/obesity and hypertension with increasing age and wealth index, similar to Bista et al. [43]. Though hypertension steadily increased with age, the prevalence of overweight and obesity inconsistently increased with age. Women in the 30–39 years age group were reported to be more overweight/obese (40.3%), whereas 40–49 years aged women had the highest prevalence of hypertension (39.2%). However, the prevalence of overweight and obesity was twice in both aforementioned age groups compared to the 18–29 years age group, which is aligned with the findings from study conducted by Khanam et al. [51]. The increasing trend of hypertension with age is also evident in another nationwide study in Bangladesh [52]. We observed education level has vice versa relation with hypertension and overweight & obesity. The hypertension prevalence was higher in the illiterate group, whereas the overweight/obesity prevalence was higher in people who had secondary education. Similar findings were described by Sun K. et al., who explained that participants with lower educational backgrounds have to involve in more physically active work than educated persons, which explained their lower rate of overweight and obesity [53]. This result also can be explained by the fact that the participants from the agriculture/self-employed group had a lower prevalence of overweight and obesity in our study. Obesity and hypertension can both endanger women's life during pregnancy, so these alarming conditions need to be addressed, and policies should be implemented.

We found that married women were more vulnerable to hypertension and prone to be obese than never-married women. Findings were supported by previous studies [43, 51]. Taking birth control methods might be an explanation for these findings [54, 55]. Similar to other nationwide studies, we found overweight/obesity and hypertension prevalence varies according to wealth index and geographical areas [51, 52]. Obesity and hypertension both significantly increased with the wealth index. This positive association with the wealth index is a clear reflection of less physical activity due to their sedentary lifestyle. Though there was no significant difference between the residential distribution for both obesity and hypertension, similar to the findings from the study of Chowdhury et al., we found the highest hypertensive participants were reported from Rangpur, and the lowest was from Sylhet. Poverty, malnutrition, and salt intake patterns might be responsible for this variation [56, 57].

We also found nearly 13% of the study participants reported having two risk factors, similar to Biswas et al.' findings [42]. We found the clustering of NCDs risk factors increased with age and wealth index. The finding of our study was aligned with the trend of increasing NCDs risk factors found in previous studies from different countries, including Bangladesh [42, 43]. Our study demonstrates that lower education is a potential covariate for NCDs risk factors. However, it was inconsistent with the findings from other population-based studies in Bangladesh [12, 58]. A specific study population with a specific age range might be reported differently in existing studies that resulted in the distribution of NCDs risk factors among the respondents. Urbanization had no significant impact on NCDs risk factors. In participants from Barishal, NCDs risk factors were reported as approximately 1.5 times more than in Dhaka. Findings are similar to the research of Al-Zubayer et. al. [58]. We assume that geographical distribution, limited health infrastructure, and human-resource capacity are hurdles that hinder disadvantaged groups of the population from receiving knowledge and preventative care, which ultimately may have an impact on the distribution of the NCDs risk factors.

Bangladesh has achieved the Millennium Development Goal-5 (MDG-5) [59]; however, the findings of this study demonstrate that reproductive-aged women are still in a vulnerable position. These findings indicate an increasing burden for NCDs among reproductive-aged women in Bangladesh. It might be a barrier to achieving Sustainable Development Goal

(SDG) [60]. Pre-existing NCDs risk factors are well-established determinants for several adverse health outcomes, including maternal death [61]. Many adverse pregnancy outcomes are significantly associated with obesity, hypertension, and the smoking status of mothers [62–64]. Taking into consideration the ages and gender of our study participants, these factors might be responsible for doubling the risk of unfavorable health conditions.

## Strengths and limitations

The major strength of our study is that we used a population-based, nationally representative data source. The BDHS covered rural and urban regions in all administrative divisions, making these findings generalizable to the country. Our modified Poisson regression corrects the overestimation of the effect size produced by conventional logistic regression employed in cross-sectional studies and increases the precision of the findings. Like the other studies, our study had some limitations as well. As we used secondary data from a cross-sectional survey, we were only allowed to find the associations between the dependent and independent variables available in the dataset. Secondly, the BDHS did not record other important clinical biomarkers (i.e., blood lipid profile, HbA1c, serum creatinine), as well as behavioral factors (e.g., alcohol consumption, sleep duration), dietary factors (e.g., type and amount of food taken), physical activity, which are crucial risk factors for NCDs. So, these data could not be included in the analyses, which limits the strength of this study.

## Conclusions

There was a high prevalence of non-communicable diseases (NCDs) risk factors among older women and who were currently married or widowed/divorced, from the wealthiest socio-economic groups, and women with less education. Furthermore, the relationship between NCDs risk factors and geographic location was found to be significant. Lessening the burden of NCDs necessitates effective disease prevention and control measures for women through reducing the burden of hypertension, overweight/obesity, and smoking. If the policymakers do not address these risk factors, women of reproductive age in Bangladesh will be at risk for poor health outcomes, and it will be difficult to reach the sustainable development goal of reducing non-communicable diseases.

## Supporting information

**S1 Data. The dataset of the study can be found in S1 Data file.**
(DTA)

## Acknowledgments

The authors thank MEASURE DHS (Demography and Health Surveys) for granting access to the BDHS 2017–2018 data.

## Author Contributions

**Conceptualization:** Saifur Rahman Chowdhury.

**Data curation:** Saifur Rahman Chowdhury, Tasbeen Akhtar Sheekha.

**Formal analysis:** Saifur Rahman Chowdhury.

**Investigation:** Saifur Rahman Chowdhury, Md. Nazrul Islam, Shirmin Bintay Kader.

**Methodology:** Saifur Rahman Chowdhury, Md. Nazrul Islam.

**Project administration:** Saifur Rahman Chowdhury, Shirmin Bintay Kader, Ahmed Hossain.

**Resources:** Saifur Rahman Chowdhury, Tasbeen Akhtar Sheekha.

**Software:** Saifur Rahman Chowdhury.

**Supervision:** Saifur Rahman Chowdhury, Ahmed Hossain.

**Validation:** Saifur Rahman Chowdhury, Md. Nazrul Islam, Tasbeen Akhtar Sheekha, Shirmin Bintay Kader, Ahmed Hossain.

**Visualization:** Saifur Rahman Chowdhury, Md. Nazrul Islam, Ahmed Hossain.

**Writing – original draft:** Md. Nazrul Islam, Tasbeen Akhtar Sheekha, Shirmin Bintay Kader.

**Writing – review & editing:** Saifur Rahman Chowdhury, Md. Nazrul Islam, Ahmed Hossain.

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
