## [Decision Letter · Decision Letter 0]

13 Sep 2022

PONE-D-22-21467Prevalence and determinants of non-communicable diseases risk factors among reproductive-aged women of Bangladesh: Evidence from Bangladesh Demographic Health Survey 2017-2018PLOS ONE

Dear Dr. Chowdhury,

Thank you for submitting your manuscript to PLOS ONE. After careful consideration, we feel that it has merit but does not fully meet PLOS ONE’s publication criteria as it currently stands. Therefore, we invite you to submit a revised version of the manuscript that addresses the points raised during the review process. Please submit your revised manuscript by Oct 28 2022 11:59PM. If you will need more time than this to complete your revisions, please reply to this message or contact the journal office at plosone@plos.org. Please include the following items when submitting your revised manuscript:A rebuttal letter that responds to each point raised by the academic editor and reviewer(s). You should upload this letter as a separate file labeled 'Response to Reviewers'.A marked-up copy of your manuscript that highlights changes made to the original version. You should upload this as a separate file labeled 'Revised Manuscript with Track Changes'.An unmarked version of your revised paper without tracked changes. You should upload this as a separate file labeled 'Manuscript'.

We look forward to receiving your revised manuscript.

Kind regards,

Mohammad Enamul Hoque, Ph.D

Academic Editor

PLOS ONE

Journal Requirements:

2. Please ensure that you refer to Figure 1 in your text as, if accepted, production will need this reference to link the reader to the figure.

Reviewers' comments:

Reviewer's Responses to Questions

**Comments to the Author**

1. Is the manuscript technically sound, and do the data support the conclusions?

Reviewer #1: Partly

Reviewer #2: Yes

2. Has the statistical analysis been performed appropriately and rigorously? 

Reviewer #1: Yes

Reviewer #2: Yes

3. Have the authors made all data underlying the findings in their manuscript fully available?

Reviewer #1: Yes

Reviewer #2: Yes

4. Is the manuscript presented in an intelligible fashion and written in standard English?

Reviewer #1: Yes

Reviewer #2: Yes

5. Review Comments to the Author

Reviewer #1: Saifur Rahman Chowdhury et al. investigated the prevalence of non-communicable diseases (NCD) risk factors among reproductive-aged women of Bangladesh. The paper is well written. Overall, the findings are more confirmatory than novel. I have a number of concerns, which are listed below:

Define reproductive age group in the abstract

The introduction section could be shortened.

Line 4: Suggest replacing the word ‘evidence’ with ‘findings’ in the title considering the results are from a survey.

Line 56-57: Please mention compared to which division the Barishal division had higher risk factors for NCD.

Line 81-84: The number of premature deaths reported 15 million globally in line 81, whereas in line 84 it was 38 million in year 2012. Suggest checking the number and report the most recent one incusing the year of report.

Line 85: Mention the year of data. Also report the total number of deaths.

Line 86: Report the NCD prevalence change over-time or actual prevalence during different time

Line 97-98: Are this prevalence among total population or certain age group.

Line 173: Check the definition used to define hypertension. The hypertension should be defined ‘SBP≥140’ OR ‘DBP≥90’ OR ‘on anti-hypertensive medication’. But in the definition provided it was ‘SBP≥140’ AND ‘DBP≥90’ OR ‘on anti-hypertensive medication’. If the later definition used to define the hypertension – redefine hypertension using correct definition and redo the analysis.

Table 3: Include whether it was univariate or multivariate analysis.

Figure 2: Suggest presenting participants with # of risk factors by age group or showing % for each age sub-group

Figure 3: include the reference group and title for x-axis

Reviewer #2: Revision of the study “Prevalence and determinants of non-communicable diseases risk factors among reproductive-aged women of Bangladesh: Evidence from Bangladesh Demographic Health Survey 2017-2018”

This study aims to investigate the Prevalence and determinants of non-communicable diseases risk factors among reproductive-aged women of Bangladesh: Evidence from Bangladesh Demographic Health Survey 2017-2018.

I have listed some specific comments below that the authors should consider before this work could be published.

Introduction:

1. It would be beneficial for the authors to include one or more references regarding the current NCD situation of reproductive women in LMIC.

2.In Page 4 line no 98, Would you kindly mention the prevalence of other NCDs?

The general observation is that there is need to consider language usage, thus the authors should not use colloquial language but statements should be supported by references to enhance authenticity of claims. This section can be improved.

Methods:

The methodology for the study is well explained. Perhaps two questions

1. One would ask is that since the BDHS used a multi stage stratified sampling, how were cluster and sample design effects dealt with in the analysis of data.

2. No mention is made of the data collection instrument – was it the stepwise instrument which will allow for comparison with other studies.

Results:

The result part for the study is well explained. The tables and the figures are well-documented and understandable.

Discussion:

This part is well written.

Limitations and implementation of the study are noted and discussed.

6. PLOS authors have the option to publish the peer review history of their article (what does this mean?). If published, this will include your full peer review and any attached files.

Reviewer #1: No

Reviewer #2: **Yes: **Sadia Afrin

---

## [Author Response · Author response to Decision Letter 0]

9 Oct 2022

Response to reviewers files attached

---

## [Decision Letter · Decision Letter 1]

11 Jan 2023

PONE-D-22-21467R1

Prevalence and determinants of non-communicable diseases risk factors among reproductive-aged women of Bangladesh: Findings from Bangladesh Demographic Health Survey 2017-2018

PLOS ONE

Dear Dr. Chowdhury,

Thank you for submitting your manuscript to PLOS ONE. After careful consideration, we feel that it has merit but does not fully meet PLOS ONE’s publication criteria as it currently stands. Therefore, we invite you to submit a revised version of the manuscript that addresses the points raised during the review process.

The manuscript is well written. However, the following remarks need to be addressed:

Smoking, highpertension, overweight and obesity are binary outcome variables. How did the authors conduct their Poisson regression analysis with binary outcome variables? Did they count the number of women in households, unions, upazilas, and districts who had NCD risk factors before performing an analysis?

As the authors described, the terms "overweight" and "obesity" have different definitions. However, they combined the two to form a single outcome variable. How is that even possible?

The authors analyzed the data in four different ways to identify the determinants for NCD risk factors: three for three risk factors and one overall. However, the results were presented only for overall determinants in the abstract.

The authors presented multivariable Poisson regression model results without presenting the corresponding univariable screening results and the criteria for variables to be included in the multivariable models.

We look forward to receiving your revised manuscript.

Kind regards,

A. K. M. Anisur Rahman, Ph.D.

Academic Editor

PLOS ONE

Reviewers' comments:

Reviewer's Responses to Questions

**Comments to the Author**

1. If the authors have adequately addressed your comments raised in a previous round of review and you feel that this manuscript is now acceptable for publication, you may indicate that here to bypass the “Comments to the Author” section, enter your conflict of interest statement in the “Confidential to Editor” section, and submit your "Accept" recommendation.

Reviewer #3: (No Response)

Reviewer #4: (No Response)

2. Is the manuscript technically sound, and do the data support the conclusions?

Reviewer #3: Yes

Reviewer #4: Yes

3. Has the statistical analysis been performed appropriately and rigorously? 

Reviewer #3: Yes

Reviewer #4: Yes

4. Have the authors made all data underlying the findings in their manuscript fully available?

Reviewer #3: Yes

Reviewer #4: Yes

5. Is the manuscript presented in an intelligible fashion and written in standard English?

Reviewer #3: Yes

Reviewer #4: Yes

6. Review Comments to the Author

Reviewer #3: (No Response)

Reviewer #4: Many countries have done and are doing NCD RF surveys as a standalone or as a part of the DHS surveys. Beyond a point there is little usefulness of prevalence and predictors of the risk factors. These are no longer research questions but more of surveillance and monitoring for change. The paper does not really add any valuable knowledge and is at best a confirmation of what is known.

DHS is largely an platform for RCH monitoring and and has added three variables related to NCDs and focuses only among women.

It would be better if a section on trends is added with maybe a table or diagram to make the paper more relevant.

Since only three risk factors are considered, clustering in this does not match the other NCDRF surveys. Clustering is not clearly defined in the paper.

Specific Suggestions:

Introduction: The para on global women status can be deleted or shortened. Only those issues covered in the paper to be retained.

Drop table 1 (as details are there in table 2) and figure 1 (not specific to this study).

Table 2 can delete one NU column by getting Hypertension column before obesity as the numbers are the same.

Limitation of causality is unwarranted as no one imputes causality in a cross-sectional study and that was never the intent of the survey.

7. PLOS authors have the option to publish the peer review history of their article (what does this mean?). If published, this will include your full peer review and any attached files.

Reviewer #3: **Yes: **Dr Valerian Mwenda

Reviewer #4: **Yes: **Anand Krishnan

<quillbot-extension-portal></quillbot-extension-portal><quillbot-extension-portal></quillbot-extension-portal>

---

## [Decision Letter · Decision Letter 2]

24 May 2023

Prevalence and determinants of non-communicable diseases risk factors among reproductive-aged women: Findings from a nationwide survey in Bangladesh

PONE-D-22-21467R2

Dear Saifur Rahman Chowdhury, 

We’re pleased to inform you that your manuscript has been judged scientifically suitable for publication and will be formally accepted for publication once it meets all outstanding technical requirements.

Kind regards,

Rushdia Ahmed, MPH, MA

Academic Editor

PLOS ONE

Additional Editor Comments (optional):

Reviewers' comments:

Reviewer's Responses to Questions

**Comments to the Author**

1. If the authors have adequately addressed your comments raised in a previous round of review and you feel that this manuscript is now acceptable for publication, you may indicate that here to bypass the “Comments to the Author” section, enter your conflict of interest statement in the “Confidential to Editor” section, and submit your "Accept" recommendation.

Reviewer #3: All comments have been addressed

Reviewer #4: All comments have been addressed

Reviewer #5: (No Response)

2. Is the manuscript technically sound, and do the data support the conclusions?

Reviewer #3: Yes

Reviewer #4: Yes

Reviewer #5: Yes

3. Has the statistical analysis been performed appropriately and rigorously? 

Reviewer #3: Yes

Reviewer #4: Yes

Reviewer #5: Yes

4. Have the authors made all data underlying the findings in their manuscript fully available?

Reviewer #3: Yes

Reviewer #4: Yes

Reviewer #5: Yes

5. Is the manuscript presented in an intelligible fashion and written in standard English?

Reviewer #3: Yes

Reviewer #4: Yes

Reviewer #5: Yes

6. Review Comments to the Author

Reviewer #3: (No Response)

Reviewer #4: No further comments. All comments have been addressed by the authors. Paper is acceptable. English might need improvement

Reviewer #5: A research study was conducted which aimed to determine the prevalence and determinants of hypertension, overweight/obesity, and smoking status. A stratified, two-stage sampling method for households was used. Poisson regression models with robust error variance were fitted to find the adjusted prevalence ratio (APR) for the factors listed above. The prevalence of hypertension, overweight/obesity, and smoking were 20.3%, 31.6% and 9.6%, respectively. Older women had more NCDs risk factors than younger women, and women without an education, married and widowed/divorced were more likely to have multiple NCDs risk factors.

Minor revisions:

1- Consistently list the factors (hypertension, overweight/obesity, and smoking) in the same order throughout the manuscript.

2- Define the abbreviation SD at its first occurrence.

3- Page 11: Provide 95% confidence intervals to describe the overall prevalence of hypertension, overweight/obesity, and smoking.

7. PLOS authors have the option to publish the peer review history of their article (what does this mean?). If published, this will include your full peer review and any attached files.

Reviewer #3: **Yes: **Dr Valerian Mwenda

Reviewer #4: **Yes: **Anand Krishnan

Reviewer #5: No

---

## [Editor Report · Acceptance letter]

2 Jun 2023

PONE-D-22-21467R2 

Prevalence and determinants of non-communicable diseases risk factors among reproductive-aged women: Findings from a nationwide survey in Bangladesh 

Dear Dr. Chowdhury:

I'm pleased to inform you that your manuscript has been deemed suitable for publication in PLOS ONE. Congratulations! Your manuscript is now with our production department. 

Kind regards, 

on behalf of

Ms. Rushdia Ahmed 

Academic Editor

PLOS ONE